# Structural basis for genome packaging, retention, and ejection in human cytomegalovirus

Zhihai Li ⬢ [1,3], Jingjing Pang ⬢ [1,2,3], Lili Dong ⬢ [1,3] & Xuekui Yu ⬢ [1,2✉]

How the human cytomegalovirus (HCMV) genome—the largest among human herpes-viruses—is packaged, retained, and ejected remains unclear. We present the in situ structures of the symmetry-mismatched portal and the capsid vertex-specific components (CVSCs) of HCMV. The 5-fold symmetric 10-helix anchor—uncommon among known portals —contacts the portal-encircling DNA, which is presumed to squeeze the portal as the genome packaging proceeds. We surmise that the 10-helix anchor dampens this action to delay the portal reaching a "head-full" packaging state, thus facilitating the large genome to be packaged. The 6-fold symmetric turret, latched via a coiled coil to a helix from a major capsid protein, supports the portal to retain the packaged genome. CVSCs at the penton vertices—presumed to increase inner capsid pressure—display a low stoichiometry, which would aid genome retention. We also demonstrate that the portal and capsid undergo conformational changes to facilitate genome ejection after viral cell entry.

[1] Cryo-Electron Microscopy Research Center, the CAS Key Laboratory of Receptor Research, Shanghai Institute of Materia Medica, Chinese Academy of Sciences, Shanghai, China. [2] University of Chinese Academy of Sciences, Beijing, China. [3]These authors contributed equally: Zhihai Li, Jingjing Pang, Lili Dong. ✉email: xkyu@simm.ac.cn

Human cytomegalovirus (HCMV), the prototypical member of the β-herpesvirinae subfamily within the *Herpesviridae*, is found in about 90% of the population worldwide[1,2]. In healthy adults and children, HCMV infection is often asymptomatic or causes mild symptoms as a consequence of host immunity. However, in immunocompromised individuals, such as patients with AIDS or transplant recipients, HCMV infection can lead to severe and sometimes life-threatening disease[3–6]. HCMV is also the leading viral cause of congenital infections that can lead to birth defects, such as intellectual disability, deafness, and even fetal death[6,7]. The virus has also recently been associated with the development of cancer[8].

HCMV has a characteristic three-layer architecture: an outer lipid bilayer envelope, an inner pseudo-icosahedral nucleocapsid, and a middle pleomorphic tegument compartment. In HCMV as well as other herpesviruses, the nucleocapsid is formed by the complex interactions of various capsid proteins. There are four major components of HCMV capsids: the major capsid protein (MCP), the triplex dimer (Tri2), the triplex monomer (Tri1), and the smallest capsid protein (SCP). During the process of HCMV infection, the nucleocapsid is delivered into host cells by membrane fusion between the viral envelope and the cell membrane. The nucleocapsid is then trafficking through the cytoskeletal system to the nucleopore, releasing the viral genome into the host nucleus in a pressure-dependent manner via the portal, which is located at a unique vertex of the nucleocapsid[9–12]. Shortly after entry, the viral double-stranded linear DNA genome is circularized and replicated via a rolling circle mechanism, leading to the formation of concatemeric intermediates of head-to-tail linked genomes[13]. These genome intermediates are recognized by viral terminase to form concatemeric DNA/terminase complex, which then binds to the portal of a preformed procapsid. The concatemeric DNA is translocated and cleaved, leaving a unit-length genome within the nucleocapsid[14–18]. Biochemical data show that DNA cleavage in herpesviruses is both sequence-specific (pac motif)[14] and pressure-dependent (head-full mechanism)[19–22].

The genome size (235 kb) of HCMV is the largest among the known human herpesviruses; albeit, the capsid size of HCMV is similar to those of other herpesviruses[23–31]. Given that the portals have been suggested to serve as head-full sensors[19–21,30,32] and that the genome sizes of herpesviruses vary considerably[33], it is still unknown how the head-full genome packaging mechanism is sensed by the portals among the different herpesviruses.

To cope with the much larger inner pressure, both pentons and hexons of the HCMV nucleocapsid are secured by a β-herpesvirus–specific pp150 tegument protein[24]; other herpesviruses, such as herpes simplex virus 1 (HSV-1) from the α-herpesvirinae, and Kaposi's sarcoma-associated virus (KSHV) and Epstein-Barr virus (EBV) from the γ-herpesvirinae, are only associated with capsid vertex-specific components (CVSCs) through their pentons[25,26,30,31]. Although biochemical data have shown that the orthologous proteins of CVSCs in HCMV are essential structural components for viral genome packaging[34], their structures and geometric distributions on nucleocapsids are still elusive.

The main functions of the herpesvirus capsid are to package, transport, and deliver the viral genome[11,35]. In accomplishing these functions, the viral nucleocapsid will encounter variable changes in the both the physical and chemical environments. For example, removal of the viral envelope and the subsequent release of non-capsid-associated tegument proteins occurs after viral cell entry. In addition, mechanical forces are imposed by the cellular motor proteins during viral nucleocapsid trafficking toward the nucleopore for genome release[9,10]. Whether the viral portal and/ or the capsid shell adopts any structural changes in response to the dynamically changing environment remains unknown.

Despite the wealth of knowledge concerning herpesviruses, there is still a lack of structural understanding as to how the large HCMV's genome is packaged, retained and ejected during viral infection. In this study, we show that the portal of HCMV is composed of a 12-fold symmetric (C12) main body, a 5-fold symmetric (C5) 10-helix anchor, and a 6-fold symmetric (C6) turret, and is specifically adapted to pack its large genome. Furthermore, we show that the copy number of CVSCs on each HCMV nucleocapsid is significantly less than that of members from the α- and γ-herpesvirinae; given that the binding of CVSCs was proposed to increase the inner capsid pressure[30], a lower stoichiometry of CVSCs would aid in the packaging and retention of this large genome into the capsid shell. In addition, the conformational changes to the portal turret and the portal-surrounding capsid proteins caused by envelope rupture of the intact virion likely represent the events that occur during viral cell entry in response to membrane fusion. Overall, our results provide insight into the conservation and adaption of HCMV portals and outline how the capsid inner pressure is sensed and modulated among herpesviruses to accomplish the tasks of genome packaging, retention, and ejection.

## Results

**Structure determinations of the in situ portal and CVSCs.** To determine the structures of the portal and CVSCs in situ, we first resolved the icosahedral reconstruction of the HCMV intact virion at 4.0 Å from 26,050 cryo-electron microscopy (cryoEM) images (Supplementary Figs. 1 and 2); the resultant reconstruction closely resembles the structure recently reported[24]. We then extracted the sub-particle images of the 12 penton vertices for each particle image using the orientation and center parameters determined in the icosahedral reconstruction. Through 3D classification and refinement with C5 symmetry applied, we identified the portal vertex and resolved its structure at a resolution of 4.2 Å, revealing high-resolution tegument densities similar to the CVSCs of other herpesviruses (Fig. 1a–c, Supplementary Fig. 2). To determine the high-resolution structure of the dodecameric portal, we re-extracted a smaller sub-particle image from each portal vertex sub-particle image. Through C5 symmetry expansion, C12 3D classification and C12 local refinement, we obtained the structure of the dodecameric portal at 4.5 Å resolution. By applying the C1 orientations of the portal sub-particle images to the corresponding images of the portal vertex sub-particles and virion particles, we then obtained the C1 reconstructions of the portal vertex and the nucleocapsid at resolutions of 5.5 Å and 6.8 Å, respectively (Supplementary Figs. 2 and 4). Both the C1 portal vertex and asymmetric capsid structures showed that the HCMV portal contains three symmetry mismatches: the dodecameric C12 main body, the C5 10-helix N-anchor, and the C6 portal turret (Fig. 1b–d and Supplementary Fig. 5). The high-resolution features revealed in the C12 reconstruction of the portal main body as well as the C5 and C6 reconstructions of portal vertex enabled us to build the near-complete atomic model of the portal protein, pUL104 (Fig. 1e and Supplementary Fig. 4). It is worth noting that 1 of the 6 coiled coils of the turret is latched by a N-terminal helix (N-latch) from one P6 MCP (Fig. 1b, c and Supplementary Fig. 5d). We designated the 6 coiled coils of the portal turret counterclockwise and numerically from the one interacting with the N-latch (1–6; Fig. 1c). The two pUL104 conformers within each coiled coil are denoted alphabetically (pUL104a and pUL104b), from the inner conformer to the outer one (inset in Fig. 1c, d).

In addition to occupying the portal vertex, the CVSCs, together with pp150, variably occupy the penton vertices. The C1 reconstruction of one-CVSC-binding penton vertex, determined

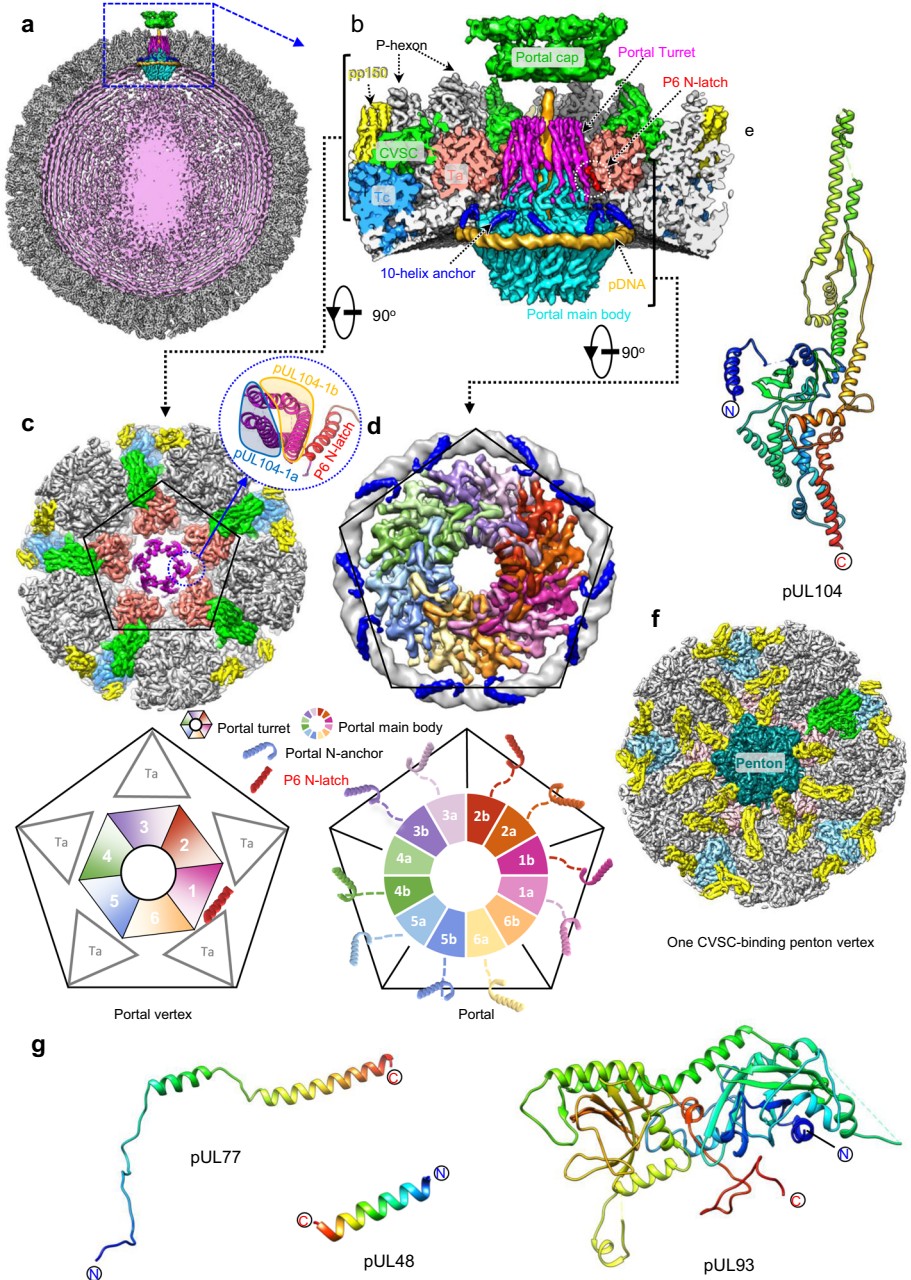

**Fig. 1 Structures of the HCMV C1 capsid, in situ portal and CVSCs. a** Central slice of the capsid asymmetric reconstruction, showing the unique portal vertex and the packaged dsDNA. **b** Zoomed-in view of the boxed region in blue in **a**, showing the interactions among portal protein, capsid proteins, and the pDNA. The capsid and tegument proteins are colored by subunit. The turret, main body, and 10-helix anchor of the portal are in magenta, cyan, and blue, respectively. **c** Top-view of the portal vertex. *Upper*, Unsharpened density map of the portal vertex. For clarity, only the turret of the portal is shown. Inset, atomic models of the boxed region, showing how one of the six coiled coils is latched by the N-terminal helix (N-latch) of one P6 major capsid protein (MCP). *Lower*, A schematic illustration of the pentagonal region from the Upper image, including the C6 portal turret (labeled 1–6), the P6 N-latch and the triplex Ta within the portal vertex. The two conformers from the coiled coil that contacts the P6 N-latch are defined as pUL104-1a (inner one) and pUL104-1b (outer one); the remainder are numerically identified counterclockwise. **d** Structural organization of the portal main body, the 10-helix anchor, and the portal-encircling (p)DNA. *Upper*, Unsharpened density maps of the portal main body (colored by molecule), the 10-helix anchor (blue), and the pDNA (gray). *Lower*, A schematic illustration of the tentative identities of the 10 conformers of the 10-helix anchor, as indicated by dashed lines. **e** Model of pUL104, colored by residue from the N-terminus (blue) to the C-terminus (red). **f** Unsharpened density map of one CVSC-binding penton vertex. The penton MCP is colored in dark cyan. The other proteins are colored as in **b**. **g** Models of the three CVSC proteins of pUL77, pUL48 and pUL93. Each protein is rainbow colored by residue from the N-terminus (blue) to the C-terminus (red).

at 4.0 Å resolution (Supplementary Figs. 2 and 3), revealed a CVSC structure essentially identical with that in the portal vertex (Fig. 1c, f): both structures are heteropentamers of two pUL77, two pUL48, and one pUL93 molecules (Fig. 1g, Supplementary Fig. 4 and Supplementary Table 1).

**Unique structure of the in situ HCMV portal**. The portal structures of several herpesviruses, such as HSV-1, KSHV and EBV, have been resolved, all of which show a similar organization: a dodecameric portal main body and an unmodeled portal turret with C5 symmetry[30,36,37]; the identities of the portal turrets

remain unknown. Our portal vertex reconstruction shows that the HCMV portal turret, consisting of six coiled coils, is clearly connected with the dodecameric portal main body (Supplementary Fig. 5). Our findings represent the first structural evidence that elucidated the portal turret is part of the portal protein pUL104. In addition, our C5 portal vertex structure revealed 10 short (27 residues) fragments of helix-loop, which are structurally similar to each other and are all located close to the portal main body (Fig. 1d). We believe these fragments are the N-terminal regions of the portal protein pUL104 based on three facts: 1) the C1 reconstruction of the portal vertex revealed that one fragment could connect to the N-terminus of the portal main body at a low threshold (Supplementary Fig. 5e); 2) the helix-loop structure is consistent with the secondary structure prediction of the pUL104 N-terminal end (Supplementary Fig. 6); and, perhaps most importantly, 3) the density map of the helix-loop fragment is in agreement with the atomic model (Supplementary Fig. 4a).

We were unable to determine the affiliations of each of the 10 N-anchors among the 12 pUL104 molecules. Thus, we tentatively assigned them to the nearest pUL104 monomer just for the convenience of structural description (Supplementary Fig. 7); our conclusions and discussions are independent of this assignment.

Based on the above analysis and by using the density maps of the C12 portal, the C5 and the C6 portal vertex reconstructions, we built the atomic model of pUL104 (Figs. 1e and 2a). We show that pUL104 consists of six domains: wing (residues 54–90, 168–193 and 239–286), crown (residues 91–167, 194–238 and 570–647), stem (residues 287–314 and 528–551), clip (residues 315–322 and 485–527), helix-rich (residues 323–338, 344–354 and 434–484) domains, as well as a β-hairpin (residues 552–569). Furthermore, it includes a novel domain, that we hereafter refer to as aileron domain (residues 13–39), because of its positional closeness to the wing domain. The wing, crown, stem, and clip domains form the C12 main body, whereas the helix-rich and aileron domains form the C6 turret and the C5 10-helix anchor, respectively (Fig. 2b, c).

The HCMV portal main body is a mushroom-shaped dodecameric complex with a central channel through which the viral genome is packaged and delivered. The central channel has three narrow regions (Fig. 2b). The first, with a diameter of 30 Å, is the tunnel structure composed of 12 sets of three-stranded β-sheets (Supplementary Fig. 8a, b). The second, with a diameter of 26 Å, is the channel valve formed by the 12 β-hairpin domains. The channel valve—the most constricted region of the portal channel—strongly holds the terminal genome DNA, and presumably functions as a check valve to prevent DNA slippage. Three interactions contribute to the stabilization of the portal main body (Supplementary Fig. 8): the first (and most important) involves extensive contacts between neighboring monomers; the second involves β-augmentation among the three-stranded β-sheet of the clip region from two adjacent subunits; and the third involves stabilization of the dodecamer portal main body through two clip helices (residues 488–493 and 495–505) from each pUL104 subunit, which simultaneously hold the stem regions of the two neighboring subunits.

Located between the portal main body and portal cap lies the third narrow region of the central channel, the portal turret, a hollow cylinder with an inner diameter of 40 Å. The turret cylinder wall comprises six coiled coils (Fig. 2b, d). The 2 helix-rich domains within each coiled coil are similar but rotated ~45° around with each other relative to the main body (Fig. 2e). The bottom opening of the turret cylinder leads to the central channel of the portal main body, whereas the top opening is loosely covered by the portal cap (Fig. 1b).

The 10-helix anchor contains 10 aileron domains arranged with 5-fold symmetry and appears as a helical ring surrounding

the portal. The C-terminal loops of the 10 aileron domains consistently point to the portal wing domains (Fig. 2c). No equivalent structures of this 10-helix anchor have been found in other viral portals.

**Interactions among the portal, capsid proteins, and the viral genome DNA.** We next identified the interactions between the portal and capsid components by docking the atomic models of the periportal capsid proteins and the portal into the C1 reconstruction of the portal vertex (Fig. 3a). We show that the turret makes contact with both the Ta triplexes and the P6 MCP. The interactions between the turret and Ta triplexes are mediated by the 10 β-hairpins from the trunk domains of five Tri1 and five Tri2B molecules, respectively (Fig. 3b). Due to symmetry mismatch, each of the 6 coiled coils of the turret interacts unequally with two β-hairpins. In addition, one coiled coil of the turret is latched by a N-terminal helix (N-latch) from one P6 MCP (Fig. 3a–c); the corresponding region in the other four P6 MCPs is flexible and not resolved (Fig. 3b).

The capsid exterior and interior are divided by the wing region of the portal main body and the capsid floor elements. The portal wing is in close contact with five sets of a long helix, a short helix of P6, and two β-sheets in the floor regions of P1 and P6 (see Fig. 3d, e). In addition, the crown region is tightly encircled by a DNA fragment, which we hereafter refer to as portal-encircling DNA (pDNA) (Fig. 3a and Supplementary Fig. 7a). pDNA seems to be a conserved element across the *Herpesviridae* family, with all known in situ portal structures of herpesviruses showing a similar ring-like DNA density wound around the portal crown region[30,36,37]. The major and minor grooves of the pDNA are clearly resolved, which enabled us to build a pseudo-model and estimate the length of pDNA as 148 bp (Fig. 3f). The pDNA is also the site of contact of the 10-helix anchor. Notably, each of the 10 helices from the portal anchor exclusively interacts with the same one of the two DNA strands, as indicated by the fact that all of the interaction sites are located on the same-side phosphate backbone with respect to the grooves of pDNA. Through hydrophobic interactions, the helix anchor is tightly held in place by the capsid floor regions of the P1 and P6 MCPs (Fig. 3g, h).

**CVSCs, together with pp150, variably occupy the penton vertex registers.** Unlike the icosahedral reconstruction, which showed that each triplex of the viral nucleocapsid was exclusively associated with three copies of pp150, our C1 reconstruction of the portal vertex revealed two types of tegument densities. The first presents with two copies of pp150 associated with the periportal triplex Tc. The other, lying on periportal triplexes Ta and Tc, is completely different but reminiscent of those of CVSCs from other herpesviruses (Fig. 1c). In addition, the asymmetric reconstruction of the HCMV intact virion shows that the tegument density above one of the five registers from both portal-proximal and portal-distal penton vertices is much weaker than that of the other four registers and could only be identified after the density map was subjected to low-pass filtering (Fig. 4). The morphology of these tegument densities is neither like a cluster of three pp150 molecules, as observed in other registers, nor like the morphology of the periportal CVSC.

Thus, to elucidate the identities of these tegument densities, we expanded the dataset of the penton vertex sub-particles with 5-fold symmetry and performed focused 3D classification with a mask that encompasses only the weaker tegument densities. We finally obtained one 3D class (13.6%) that showed a tegument density similar in both strength and morphology to that of the periportal CVSC (Supplementary Fig. 2). Notably, the 13.6%

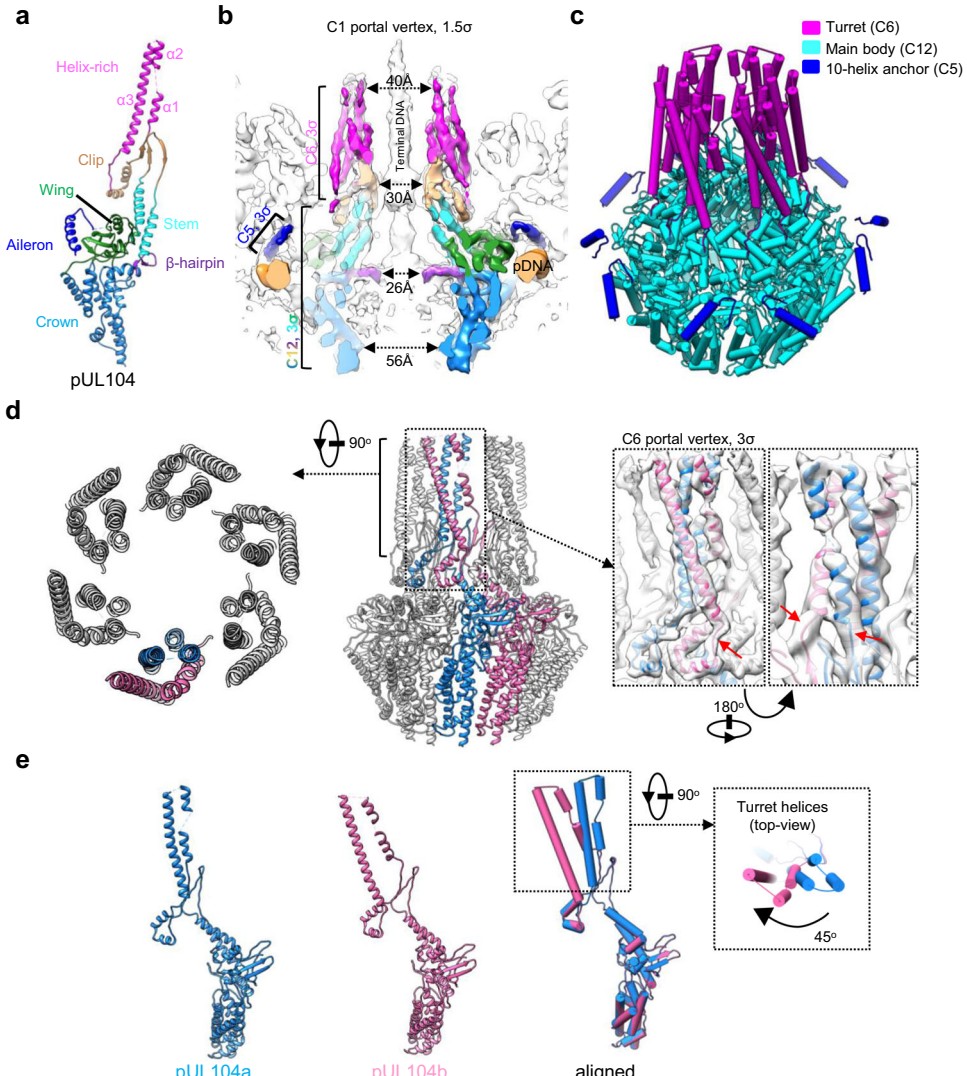

**Fig. 2 In situ structure of the portal. a** Atomic model of pUL104 portal protein, colored by domain. **b-c** Structure of the portal. **b** Superposition of the C1 density map of the portal vertex region (transparent gray) with the segments of the C6 portal turret, the C12 portal main body, the C5 portal 10-helix anchor and the pDNA (orange). The portal protein is colored as in **a**. The contour levels of all protein density maps are indicated. **c** The pipe-and-plank depiction of the portal. The C6 turret, the C12 main body, and the C5 10-helix anchor are in magenta, cyan, and blue, respectively. **d** Atomic model of the portal turret and main body. Left, top view of the portal turret, showing the six coiled coils. Right, side view of the portal turret and main body. Two conformers of the 12 pUL104 molecules are highlighted in blue (pUL104a) and pink (pUL104b), respectively. *Inset*, C6 density map of the portal vertex and the model of the boxed region, showing the density connection between the turret and main body, as indicated by the red arrows. **e** Atomic models of the two pUL104 conformers of pUL104a (left) and pUL104b (middle). Superimposition of the two conformers (right and inset) shows that the 2 helix-rich domains within each coiled coil are similar but rotated about 45° with each other relative to the main body.

CVSC occupancy of the peripenton registers in HCMV is significantly lower than that in HSV-1 (100%)[37], KSHV (37.9%)[36], and EBV (20.3%)[30]; albeit, HCMV contains the largest DNA genome. Indeed, the CVSC copy number associated with the nucleocapsid among different herpesviruses seems to be inversely proportional to genome size, further supporting the suggestion that the peripenton CVSCs should have no contribution to the overall stability of the nucleocapsid[30]. Given that each HCMV nucleocapsid has 5 periportal and 7-8 peripenton CVSCs ($13.6\% \times 55 \approx 7.5$), and that each peripenton and periportal CVSC would replace 3 and 4 copies of pp150, respectively (Fig. 5a, b), we calculated that each HCMV nucleocapsid has 12–13 copies of CVSCs and 916–919 copies of pp150s on average.

We next analyzed the repeat numbers and geometric relationships of the sub-particles in the data files of the C1 focused classification. We identified different penton vertices with respect to the CVSC-binding patterns: one-CVSC-binding (45.94%), meta-CVSC-binding (two adjacent registers were occupied, 4.78%), ortho-CVSC-binding (two registers with a gap in between were occupied, 4.12%), meta-CVSC-absent (0.38%), ortho-CVSC-absent (0.35%), four-CVSC-binding (0.033%), five-CVSC-binding (0.0017%), and CVSC-absent (44.92%) penton vertices. We obtained the reconstructions for most of the other penton vertices, except for the four- and five-CVSC-binding penton vertices, of which there were too few for reconstruction (Supplementary Fig. 2).

**Structure of the CVSCs and their interactions with the capsid proteins**. The heteropentameric CVSC of HCMV contains one pUL93 (467 of 594 residues), two pUL77 N-terminal regions (residues 2–81 for the upper one and residues 12–77 for the lower one) and two pUL48 C-terminal helices (residues 2221–2240)

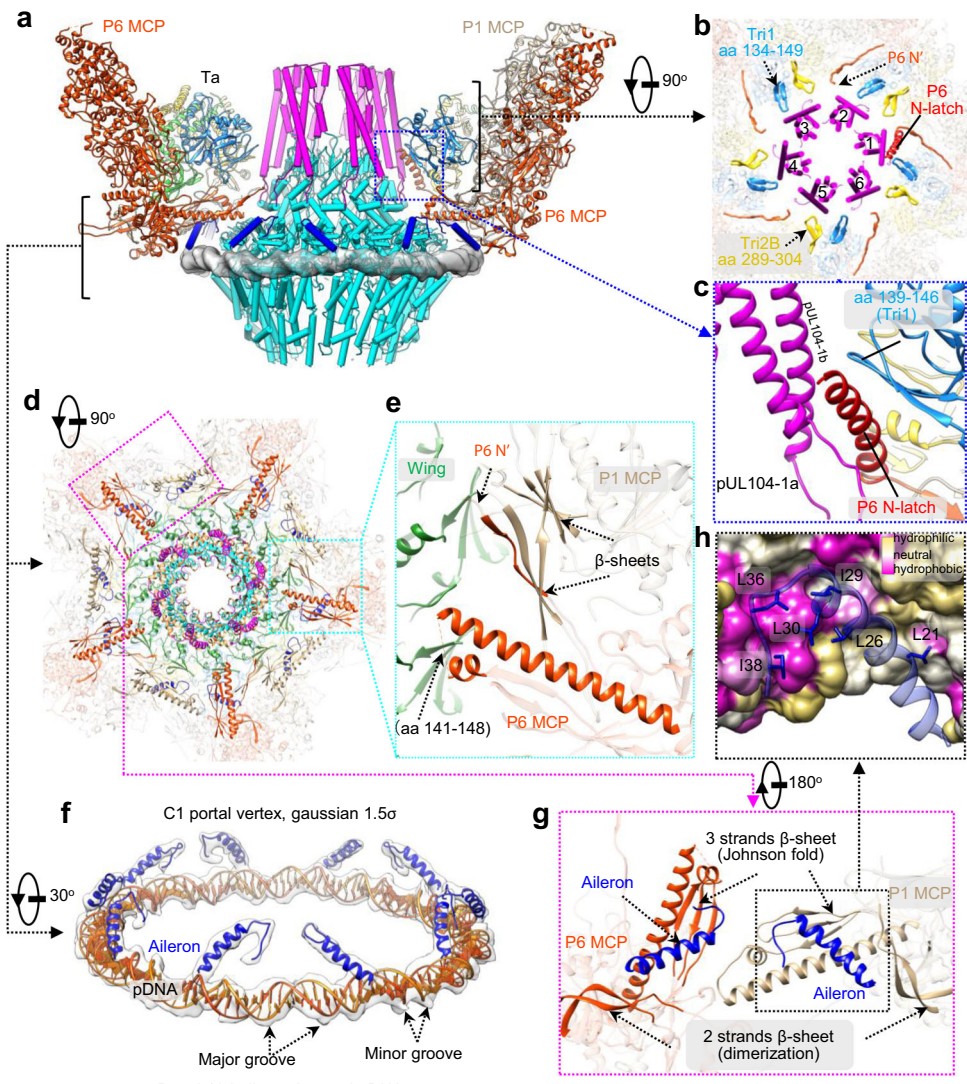

**Fig. 3 Interactions among the portal, the capsid proteins and the genome. a** Side view of the portal vertex. For clarity, only the P1 MCP, two P6 MCPs, and two triplexes Ta are shown. The Tri1, Tri2A and Tri2B are in light blue, green, and yellow, respectively. The turret, main body, and 10-helix anchor of the portal are in magenta, cyan, and blue, respectively. The pDNA density is in transparent gray. **b** Top view of the upper region of the portal vertex as indicated in **a**, showing the symmetry-mismatched interactions between the C6 portal turret and the C5 capsid proteins. The N-terminal regions of the five P6 MCPs are in orange-red; the N-latch from one of these is indicated in red. The regions of Ta involved in interactions with the portal turret are indicated. **c** Enlarged view of the boxed region in **a**, showing the P6 N-latch contacting both the portal turret and a β-hairpin of Tri1. **d** Top view of the lower region of the portal vertex as indicated in **a**, showing the interactions between the main body and the 10-helix anchor of the portal and capsid proteins. **e** Enlarged view of the boxed region in cyan in **d**, showing the interactions between the portal wing and one of the five sets of a long helix and a short helix in the P6 Johnson-fold, the two β-sheets contributed by the P1 Johnson-fold domain, and the P6 N-terminal region. **f** C1 reconstruction (transparent gray) and models of the portal 10-helix anchor (blue) and the pDNA (orange). The major and minor grooves featuring typical duplex DNA are indicated. **g–h** Bottom view of the boxed region (magenta) in **d**, showing the interactions between the portal aileron domain and the capsid floor region. Two conformers of the portal aileron have quasi-equivalent contacts with the capsid floor region formed by P6 and P1 MCPs, respectively (**g**), mainly mediated through hydrophobic interactions (**h**). The hydrophobic residues from the portal aileron domain are highlighted by showing the side chains. The capsid proteins are shown in hydrophobic surface representation.

(Fig. 6a–c). The pUL93 comprises three domains: front (residues 1–50 and 89–345), back (residues 51–88 and 419–591) and top (residues 346–418) domains (Fig. 6b). The two pUL77 N-terminal regions have similar structures, consisting of an N-terminal domain (residues 2–29 for the upper one and residues 12–29 for the lower one) and a helix domain (residues 30–81 for the upper one and residues 30–77 for the lower one). The helix domains of the two pUL77 molecules make contact with the two pUL48 C-terminal helices to form a four-helix bundle that sits above the front domain of the pUL93 molecule. In contrast, the N-terminal

domains of the two pUL77 molecules interlace with each other, as featured by two β-augmentations (Fig. 6c). The interlaced N-terminal domains are then tightly fastened to the underlying pUL93 through two prominent interactions: 1) one two-stranded β-sheet of pUL77 is augmented by the β-strand of the pUL93 top domain to form a three-stranded β-sheet; 2) the two helices (residues 363–382 and residues 400–415) from the top domain of pUL93 are joined by the helix (residues 30–42) of the N-terminal domain from the lower pUL77 to form a three-helix bundle (Fig. 6c).

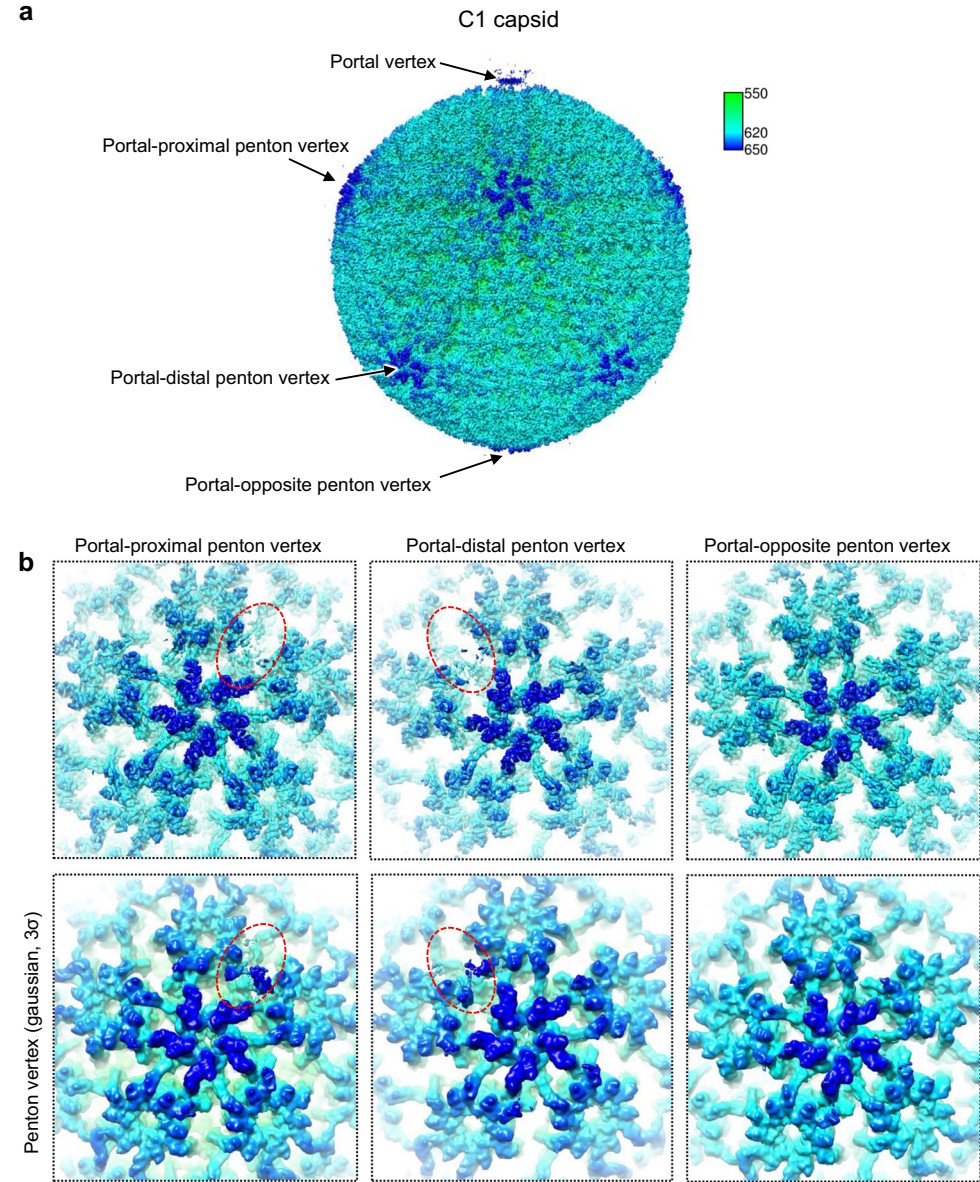

**Fig. 4 C1 reconstruction of the virion capsid. a** Radially colored surface representation of the C1 reconstruction of the virion capsid. The color key is indicated. **b** Zoomed-in views of the three types of penton vertices, the portal-proximal, the portal-distal and the portal-opposite penton vertices, showing the disordered densities (red-dashed ovals) at one of the five pp150 registers for both the portal-proximal and the portal-distal vertices. *Upper*, Original maps. *Lower*, Gaussian-filtered maps.

Previous work shows that α- and γ-herpesviruses have two globular densities[30,36,37] contributed by the head domains of the homologs of the two pUL77 conformers. For HCMV, the CVSC shows only one globular density located on the left side of the four-helix bundle (Fig. 6d, e). We assigned globular density to the head domain of one of the two pUL77 molecules. This globular density clearly shows secondary structure elements in the Gaussian-filtered map, which enabled us to build a homology model of the pUL77 head domain using the crystal structure of the HSV-1 pUL25 head domain (the pUL77 equivalent, PDB: 2F5U[38]) as a reference (Fig. 6e). Instead of contacting with the penton, as observed in members of the α- and γ-herpesvirinae, the head domain of the pUL77 interacts only with the P2 hexon. Specifically, two helices of the head domain contact the SCP and the upper domain of P2 MCP, respectively (Fig. 6e).

Each CVSC binds to both triplexes Ta and Tc. Compared with the CVSC-absent (pp150 only) Ta, the CVSC-binding Ta is

rotated ~120° counterclockwise (Fig. 6f). Accompanying this Ta rotation, a curved loop region (residues 1143–1165) of the P6 MCP, which contacts the right side of Tri2B of the unrotated Ta, is transformed into an extended loop-helix-loop region, with the helix tip (residues 1145–1151) inserting into the cleft formed by the Tri1 third-wheel domain and the Tri2A clamp domain (Fig. 6f). As such, the configuration of the rotated Ta would be firmly fixed to facilitate CVSC binding. In adapting to the rotation, the lower end of the remaining pp150 (associated with the rotated Ta) undergoes a slight conformational change to interact with Tri2B; this is as compared with the pp150 counterpart, which makes contact with the Tri2A of the unrotated Ta (Supplementary Fig. 9).

The interaction between the CVSC and the rotated Ta is mediated mainly by two anchor regions (AR) of pUL93 (Fig. 7a). The first is the AR1 loop-helix region (residues 297–310) from the pUL93 front domain (Fig. 6b): the helix of

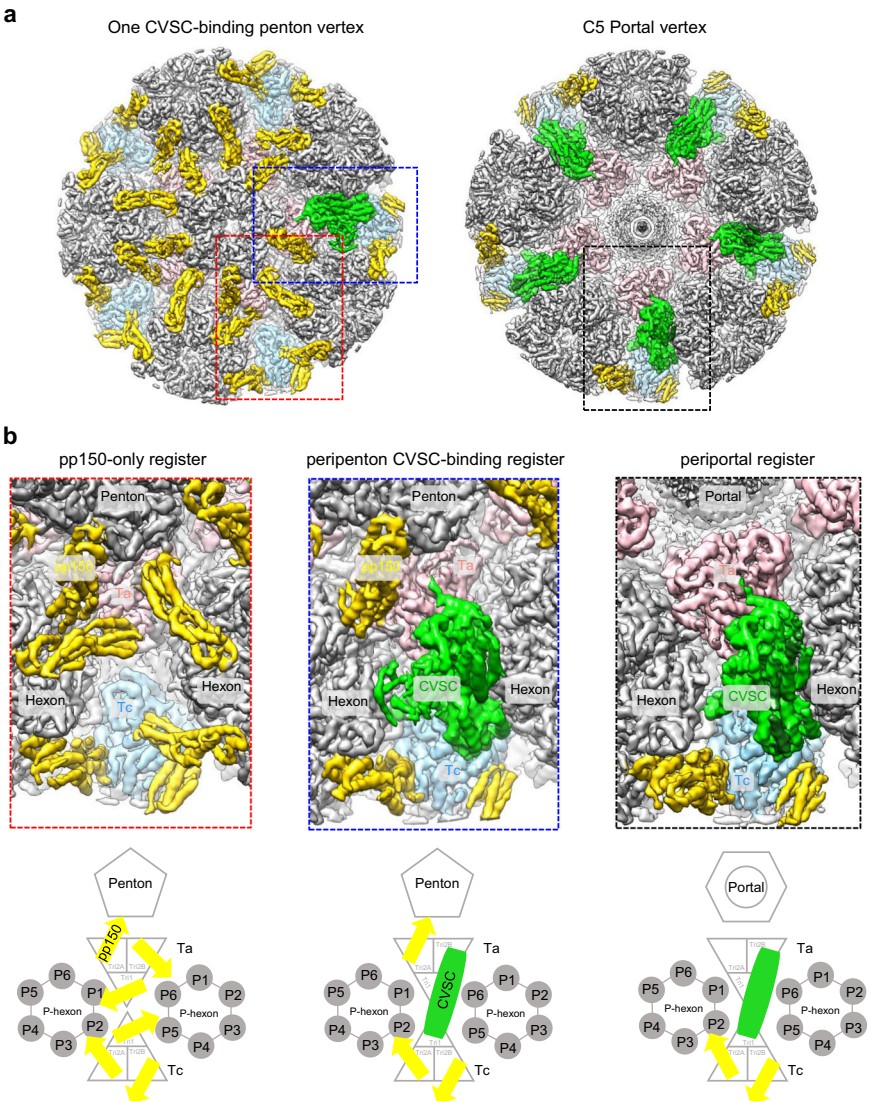

**Fig. 5 Structural comparison of pp150-only, peripenton CVSC-binding, and periportal registers of HCMV. a** Unsharpened maps of the CVSC-binding penton vertex (left) and the portal vertex (right). The capsid penton and hexons, and the disordered portal are in gray. The triplexes Ta and Tc, pp150, and CVSC are in pink, light blue, yellow, and green, respectively. **b** Zoomed-in views of the boxed regions in **a**, showing the arrangements of pp150 and the CVSC at the three different registers: pp150-only (red box), peripenton CVSC-binding (blue box), and periportal registers (black box). *Upper*, Density maps of the boxed regions in **a**. *Lower*, Schematic illustrations of the protein arrangements of the three registers.

AR1 is accommodated into a shallow cleft formed by Tri1 and Tri2B, with the loop fragment contacting Tri2B and Tri2A (Fig. 7b). The second is the AR2 loop region (residues 562–591) in the pUL93 back domain (Fig. 6b): this loop twists and turns along the surface of Tri2A, passes through a groove of Tri2A, and extends and contacts Tri1 (Fig. 7c).

The interaction between the CVSC and Tc is mediated by the N-terminal two-stranded β-sheet of pUL77 and the AR3 region (residues 450–464) in the pUL93 back domain (Fig. 7d–f). The β-sheet contacts with the embrace domain of Tri2B (Fig. 7d), while the helix of AR3 simultaneously contacts three helices to form a four-helix bundle: one helix (residues 110–119) from Tri1 and two (residues 1144–1152 and 1246–1253) from P2 MCP (Fig. 7e, f). In addition, the CVSC also contacts with P6 MCP. The featuring CVSC four-helix bundle interacts with both the upper domain of P6 MCP and the associated SCP of P6 MCP (Fig. 7g), while one helix of the upper pUL77 contacts with the upper domain of P6 MCP (Fig. 7h).

**Conformational changes caused by rupture of the viral envelope.** All herpesviruses enter host cells through membrane fusion, which is mediated by viral glycoproteins[39]. Given that the nucleocapsids of herpesviruses are highly pressurized[24], rupture of the virion envelope and the subsequent release of the non-capsid-associated tegument proteins during viral cell entry would cause conformational changes of to the nucleocapsids. To explore this hypothesis, we obtained partially-enveloped nucleocapsids by treating HCMV intact virions with detergent immediately before cryoEM sample preparation (Supplementary Fig. 10). Using the same data processing strategy for that of the HCMV intact virion, we sequentially resolved the structures of the C5 portal vertex, the C12 portal, the C1 portal vertex, the C6 portal vertex and the C1 capsid of the partially-enveloped HCMV nucleocapsid at resolutions of 3.9 Å, 4.2 Å, 4.8 Å, 5.3 Å, and 6.3 Å, respectively (Supplementary Fig. 11 and Supplementary Table 1). Compared with the intact virion, the portal vertex of the partially-enveloped nucleocapsid underwent substantial conformational changes.

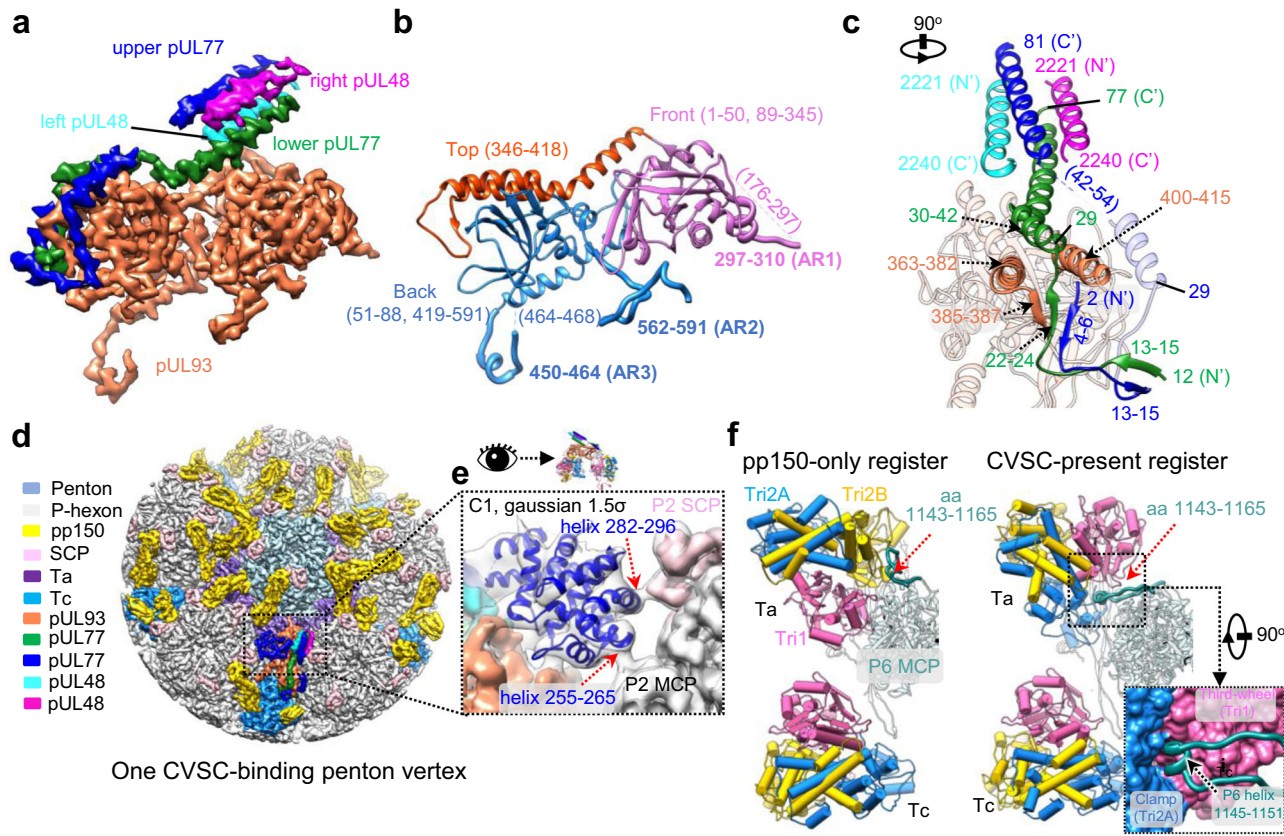

**Fig. 6 The CVSC structure. a** Sharpened density map of CVSC, colored by protein. **b** Model of pUL93, colored by domain, as indicated. Three anchor regions (ARs) are highlighted by a thicker ribbon model. **c** Back view of the CVSC atomic model. The secondary structural elements involved in the molecular interactions are highlighted and indicated. **d** C1 reconstruction of one CVSC-binding penton vertex. The color key for the different proteins is indicated. **e** Zoomed-in view of the boxed region in **d**. The C1 colored map was superimposed with its Gaussian-filtered map (1.5σ), with the head domain of pUL77 replaced by the homology model. The contacting regions of the pUL77 head domain with the MCP and SCP are indicated by red arrows. **f** The atomic models of Ta, Tc, and P6 MCP in the pp150-only (left) and the CVSC-binding (right) registers. The Ta in the CVSC-binding register rotates about 120° counterclockwise as compared to that in the pp150-only register. A curved loop of P6 MCP in the pp150-only register transforms into a loop-helix-loop in the CVSC-binding register. *Inset,* Zoomed-in view of the boxed region in **f** right, showing the helix (residues 1145–1151) of the P6 MCP inserting into the cleft formed by the Tri1 third-wheel domain and the Tri2A clamp domain.

First, the portal turret was no longer latched by the N-latch of the P6 MCP (Fig. 8a) and became largely disordered (Fig. 8b); albeit, the remaining turret densities still indicated some semblance of C6 symmetry (Fig. 8b). Second, while the portal vertex, including the triplexes, the MCPs, pp150 and the CVSCs, expanded outward and away from the symmetric axis using the portal-distal end of MCP floor region as a pivot point, the portal moved upwards about 1.9 Å (Fig. 8c and Supplementary Fig. 12). In addition, MCP and Ta showed some local structural changes in the partially-enveloped capsid. The trunk loop (aa 136–146) of Ta, Tri1, and part of the long helix of the P6 MCP Johnson-fold—involved in interactions with the portal turret and wing of the intact virion, respectively—became flexible (Fig. 9a–c). By comparison, the portal cap remained essentially unchanged (Supplementary Fig. 13), further supporting the genome-securing function of the portal cap[24,30,36].

## Discussion

Using cryoEM, we resolved the in situ structure of the portal from intact HCMV virions. The portal consists of a C6 turret, a C12 main body and a C5 10-helix anchor (Fig. 2c). The C12 main body of the portal resembles that observed in other herpesviruses, such as HSV-1[37], KSHV[36], and EBV[30], and is tightly encircled by a fragment of genome DNA, which we referred to as portal-encircling DNA (pDNA; Fig. 1b). Previous studies show that, in

bacteriophage P22, the portal in the virion is also tightly wound by dsDNA but adapts a conformation that is significantly different from that of the isolated portal[40]. Lander and colleagues proposed that the P22 portal was responsible for sensing and transducing the head-full signal through these structural changes, likely caused by the tight dsDNA spooling[40]. Interestingly, the portal of the EBV virion has a much more compact conformation than that of the free portal assembled in vitro[30]. In addition, the portal of the HSV-2 virion is translocated outwards by ~30 Å as compared with that of the B capsid, a type of scaffold-containing capsid that fails to initiate DNA packaging[41]. It is likely that the portals of herpesviruses sense the pressure through conformational changes and outward movements of the portal, which are caused by the tightening of the pDNA and the increased capsid inner pressure as the genome package proceeds. Once the portal of the herpesvirus is transformed into its most compact state and has arrived at its outmost position, the genome package has reached a head-full state.

The genome of HCMV is much larger than that of members of the α- and γ-herpesvirinae, and thus is more tightly packaged within the capsid[24,42]. Accordingly, our portal structure of HCMV shows two prominent differences from that of other herpesviruses—the 10-helix anchor and the portal turret (Supplementary Fig. 14)—which we believe are related to how the larger HCMV genome is packaged and retained. The 10-helix

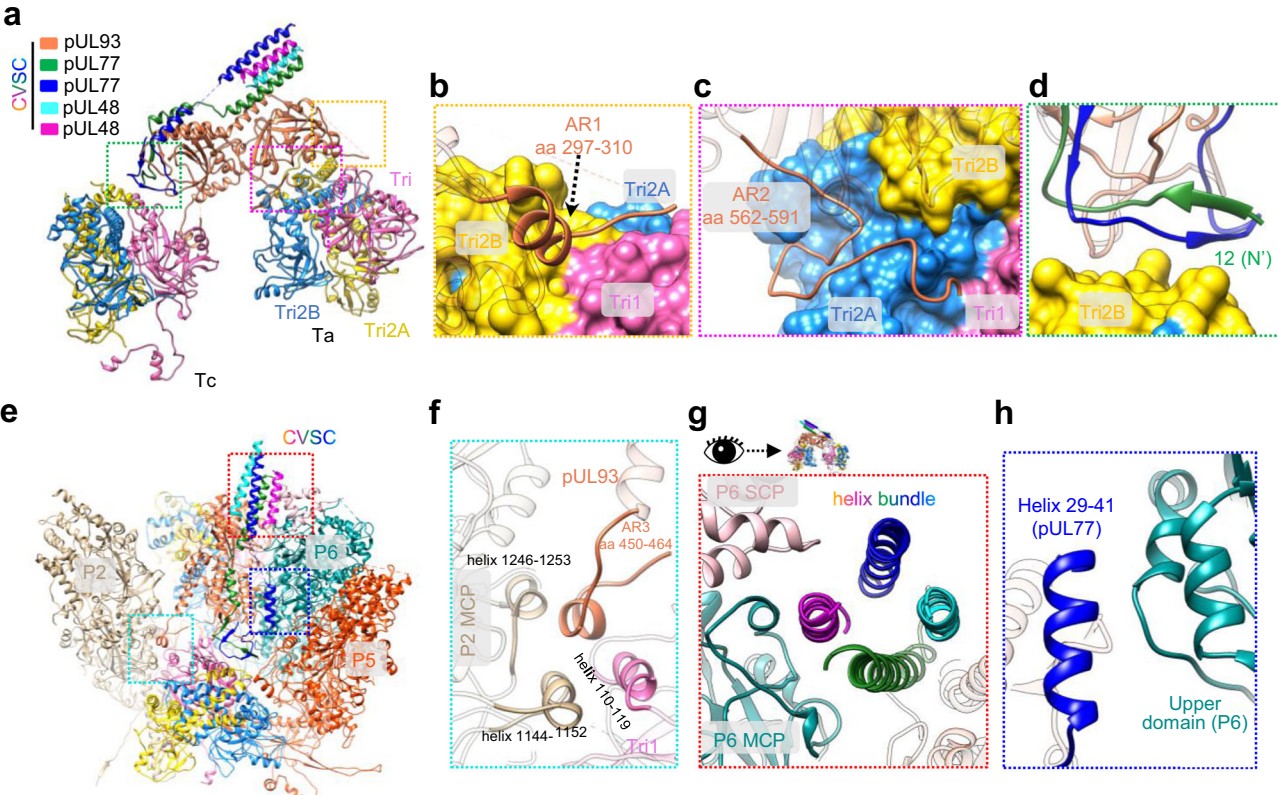

**Fig. 7 Interactions between CVSC and the capsid proteins. a** Atomic models of the CVSC and the underlying Ta and Tc, colored by molecule. **b**, **c** Zoomed-in views of the boxed regions in **a**, showing the interactions between the CVSC and Ta via anchor region 1 (AR1) (**b**) and AR2 (**c**) and Tc (right), respectively. **d** Zoomed-in view of the green boxed regions in **a**, showing the interaction between the CVSC and Tc via the N-terminal two-stranded β-sheet of the pUL77 molecules. **e** Atomic models of CVSC and its contacting capsid proteins, including Ta, Tc, and MCPs of P2, P5 and P6. The model is colored by molecule, as indicated. **f** Zoomed-in view of the blue boxed regions in **e**, showing the four-helix bundle contributed by AR3 in pUL93, Tri1, and P2 MCP. **g** Zoomed-in views of the boxed regions in **e**, showing the interactions between the four-helix bundle of CVSC and the P6 MCP. **h** Zoomed-in views of the boxed regions in **e**, showing the interactions between a short helix (residues 29–41) from pUL77 of CVSC and the P6 MCP.

anchor is not found in any other known portals. Each of the 10 fragments of the 10-helix anchor is strongly associated with the capsid floor, and interacts with the pDNA (Fig. 3f–h). The pDNA is a conserved DNA segment that is found in different herpesviruses, and presumably functions to squeeze the portal as the genome is pumped into the capsid. We surmise that perhaps this 10-helix anchor—unique to HCMV—functions to dampen this squeezing/tightening process. Consequently, a higher inner capsid pressure would then be required for the HCMV portal to reach a headful state. The second prominent structure difference in the HCMV portal is the portal turret. Unlike other herpesviruses that show 5-fold symmetry, the turret of the HCMV portal contains six coiled coils arranged with 6-fold symmetry. The turret of the portal in the intact virion is specifically strengthened through one of these coiled coils that is latched by the N-terminal helix of the P6 MCP. This interaction may help to retain the packaged genome.

Herpesvirus genome packaging[19–21] and ejection[11,12] processes are pressure dependent. Previous work[30] shows that CVSC occupancy at the penton vertices of HSV-1, KSHV, and EBV is inversely proportional to genome size. This relationship led to the proposal that CVSCs are present to increase the inner capsid pressure and facilitate a balance between genome retention and ejection[32]. This is achieved through stoichiometric binding to the penton vertex of capsid. This hypothesis is further supported by the results of our study. We find that, compared with other herpesviruses, HCMV contains the fewest CVSCs but has the largest genome; Thus, we surmise that the lower stoichiometry of

CVSCs in HCMV could be to encourage a higher degree of genome packaging and retention. Furthermore, it is noteworthy that the nucleocapsid of HSV-1 virion contains the scaffold proteins[43,44], the existence of which will increase the nucleocapsid inner pressure to facilitate the headful sensing and the genome ejection. However, the HCMV virion eliminates the scaffold proteins[44–47]; this also contributes to the accommodation of the larger genome.

All herpesviruses enter the host cell through membrane fusion to release the nucleocapsid. Our investigations with the partially-enveloped capsid showed that the portal and its surrounding capsid proteins undergo conformational changes once the virion envelope is ruptured. On the one hand, the outwards movement (Fig. 8c) of the portal vertex would alleviate the inner capsid pressure to facilitate genome retention during nucleocapsid trafficking. On the other hand, the disassociation of the portal turret with the MCP N-latch and the subsequent structural disorder (Fig. 8a, b) likely prime the nucleocapsid for genome ejection. These results suggest that the balance between genome retention and ejection is a dynamic process in the viral life cycle.

## Methods

**HCMV virion preparation**. Cell culture, viral growth and purification were carried out as described previously[24]. In brief, human fibroblast MRC-5 cells were grown in Eagle's Minimal Essential Medium (EMEM) supplemented with 10% fetal bovine serum (FBS), and were incubated in a humidified incubator with a 5% $CO_2$ at 37 °C. At 80% confluence, the cells were infected with HCMV strain AD169 at a multiplicity of infection (MOI) of 0.1. Twenty-four hours after infection, the culture medium was replaced with fresh medium containing 10% FBS. At 12 days'

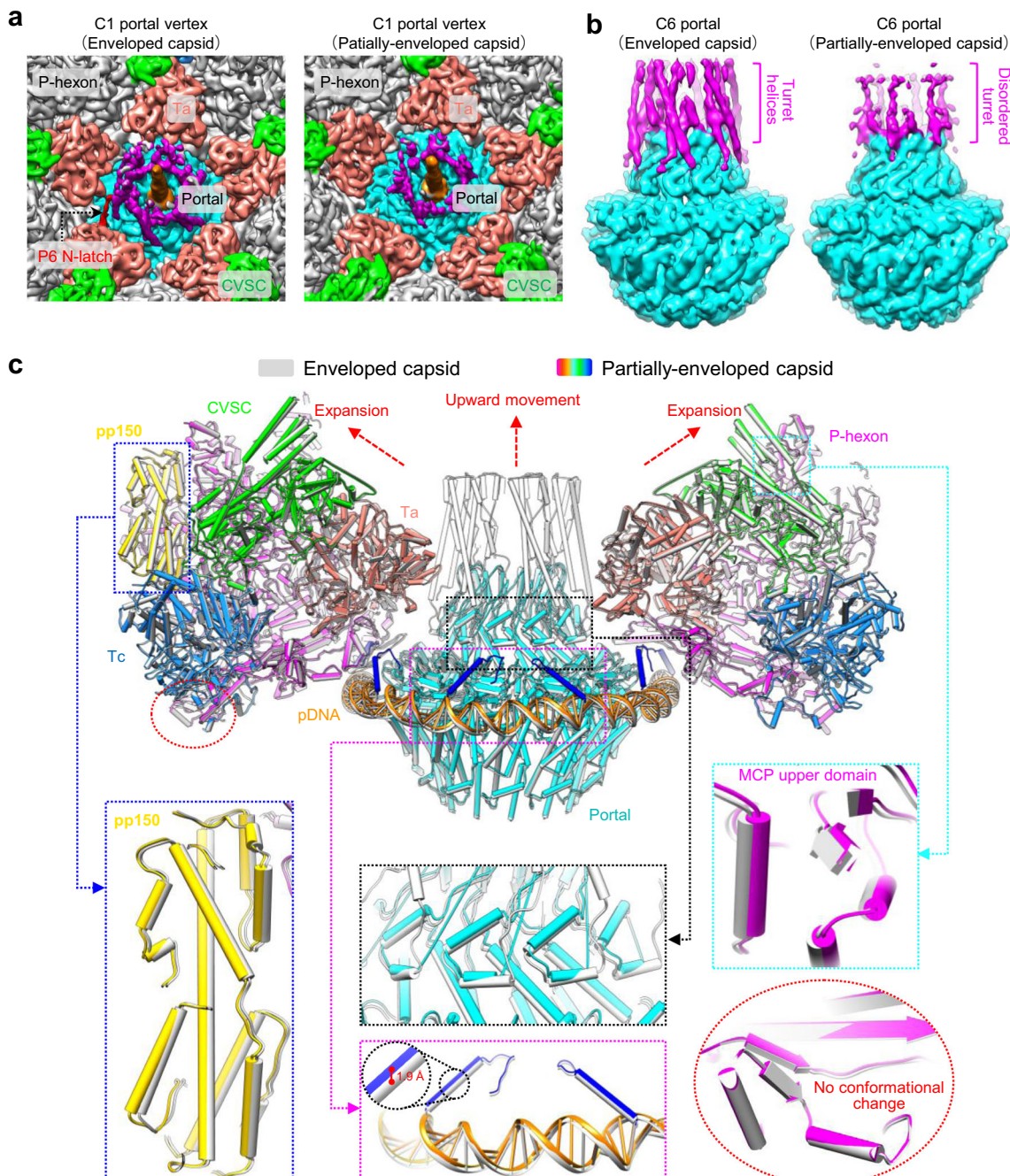

**Fig. 8 Structural comparison of the portals from the intact virion and the partially-enveloped capsid. a** C1 density maps of the portal vertex regions, showing that the portal turret and the P6 N-latch in the intact virion (left) are largely disordered in the partially-enveloped capsids (right). Capsid proteins are colored as indicated, with the P6 N-latch highlighted in red. The turret and the main body of the portal are in magenta and cyan, respectively. The terminal genome DNA is in orange. **b** C6 density maps of the portals from the intact virion (left) and the partially-enveloped capsid (right). The turret and main body of the portal are in magenta and cyan, respectively. **c** Superimposition of the portal vertices from the intact virion (gray) and the partially-enveloped capsid (colored by molecule). For clarity, only two opposing sets of the Ta, the Tc, the CVSC, and the P-hexon of the capsid proteins are shown. *Insets*, Show the conformational changes to the portal vertex caused by the breakage of the virion envelope.

post-infection, when ~70% of the cells were lysed, the culture media was collected and centrifuged at $10,000 \times g$ for 12 min to remove the cell debris. The supernatant was collected and centrifuged at $80,000 \times g$ for 1 h to pellet the viral particles. The pellet was resuspended in phosphate-buffered saline (PBS, pH 7.4) and then further purified by centrifugation through a 15–50% (w/v) continuous sucrose gradient at $75,000 \times g$ for 1 h. The light-scattering band containing the viral particles was collected, diluted with PBS to a volume of 13 mL, and pelleted by centrifugation at $75,000 \times g$ for 1 h. The pellets were finally resuspended in 12 μL of the PBS.

**CryoEM sample preparation and data acquisition**. Two cryoEM grids of intact virion were prepared by applying aliquots of 2.5 μL viral sample to a glow-discharged 300-mesh Quantifoil grid (R1.2/1.3), which were then blotted with filter paper for 14.0 s and frozen by plunging into liquid ethane using an FEI Vitrobot IX. To prepare partially-enveloped nucleocapsid samples, immediately before preparation of the cryoEM grids, the viral samples from above were mixed with Triton X-100 to a final concentration of 1.2% to disrupt the envelop of the intact virions. These partially-enveloped nucleocapsid samples were then frozen using the same way for intact virion sample.

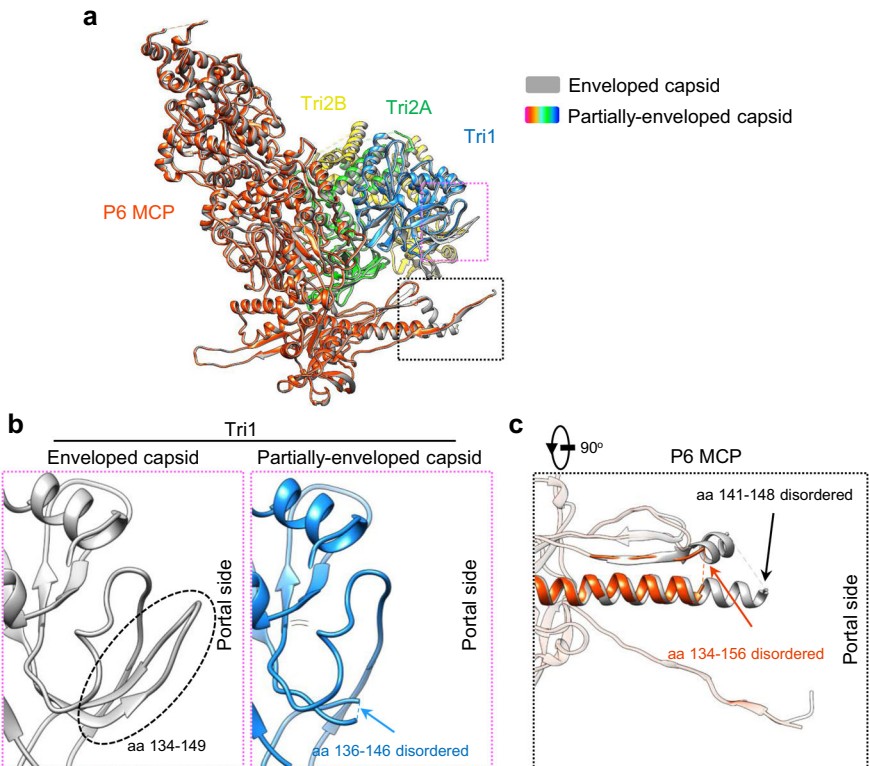

**Fig. 9 Conformational changes of the P6 MCP and Ta with the portal vertex caused by rupture of the virion envelope. a** Superimposition of the P6 MCPs and the triplexes Ta from the intact virion (gray) and the partially-enveloped nucleocapsid (colored by molecule). **b** Zoomed-in view of the magenta boxed region in **a**, showing that the trunk loop (aa 136–146) in the intact virion (left) become disordered in the partially-enveloped nucleocapsid (right). **c** Zoomed-in view of the black boxed region in **a**, showing that part of the long helix region of the Johnson-fold becomes disordered in the partially-enveloped capsid.

CryoEM micrographs for both samples were collected on a 300 kV Titan Krios microscope (FEI) equipped with a Gatan Imaging filter (GIF) and a K3 direct electron detector in super-resolution mode. The microscope was operated at 300 kV with a nominal magnification of 53,000×, yielding a calibrated pixel size of 0.8125 Å on specimen. Using the software package SerialEM[48], a total of 11,772 and 11,024 movies were collected for the intact virion and the partially-enveloped nucleocapsid samples, respectively, at a dose rate of 10 electrons/Å²·s for 3 s.

**CryoEM image processing and icosahedral reconstruction of virion capsids.** For each movie stack, 36 frames were aligned by beam-induced motion correction with the program MotionCor2[49], and dose-weighted frames with 2 time-binning in each stack were used for further processing. The defocus values and astigmatism parameters for each micrograph were determined by CTFFIND4[50]. Well-separated HCMV virion particles were picked manually using the Manual Picking function in Relion 3.0[51]. Icosahedral reconstruction of the HCMV virion capsid was performed with Relion 3.0, following the procedures previously described[30]. Briefly, the 8 time-binned particle images (256 × 256) were extracted and subjected to 2D and 3D classifications. A total of 26,050 particles from a good 3D class were selected and re-extracted with 2 time-binning (1024 × 1024) for 3D refinement. A cryoEM structure of the HCMV nucleocapsid at 4.4 Å resolution was obtained by conventional icosahedral reconstruction. After the Ewald-sphere curvature correction, we obtained a 4.0 Å icosahedral reconstruction.

**Sub-particle refinement of the portal-vertex and portal dodecamer, and asymmetric reconstruction of the virion capsid.** As illustrated in Supplementary Fig. 2, the structures of the non-icosahedral symmetric elements—the portal vertex, portal, virion capsid, and one CVSC-binding penton vertex—were determined as previously described[30]. Briefly, the icosahedral orientations and center parameters determined above were used to locate and extract the 12 vertex sub-particles of each capsid with a Scipion plugin Localized_Reconstruction[52]. Orientations and centers of all the vertex sub-particles were refined through a round of 3D refinement. The refined vertex sub-particles were then subjected to a round of focus alignment (C5). Two of the six converged classes, responsible for 7.9% of the sub-particle dataset, showed prominent portal features, whereas the other four classes displayed a penton at the center. We removed the redundant sub-particles according to the flag _rlnMaxValueProbDistribution and yielded a portal vertex

dataset that included 23,136 sub-particles. Finally, a C5 reconstruction of the portal vertex at a resolution of 4.2 Å was obtained.

To determine the high-resolution structure of the dodecameric portal, we expanded the dataset of portal vertex with 5-fold symmetry, further extracted the sub-particles only covering the lower portal part of portal vertex, and then performed a round of 3D classification (C12) without rotation alignment. One of the six converged classes, accounting for 1.0% of the dataset, failed to present any structural features, whereas the other five classes showed a dodecameric portal structure encircled by DNA containing 19.9%, 19.8%, 20.0%, 19.8%, and 19.3% of the symmetry-expanded sub-particle dataset, respectively. After removing the redundant particles in the class with a ratio of 20.0%, we finally obtained a total of 22,087 portal sub-particles. By imposing C12 symmetry, we refined the dodecameric reconstruction of the portal to a resolution of 4.5 Å. After applying the orientation parameters (_rlnAngleRot, _rlnAnglePsi and _rlnAngleTilt) of the above 22,087 portal sub-particles to their corresponding portal-vertex sub-particles and capsid particles, the asymmetric reconstructions of the portal vertex and the capsid were determined at global resolutions of 5.5 Å and 6.8 Å, respectively.

Finally, to enhance the density of the portal turret, we reconstructed the C6 structure of the portal vertex with C6 symmetry imposed to both half maps. The final resolution of the C6 portal vertex was determined to a resolution at 5.9 Å.

The global and local resolutions for all reconstructions were determined by Fourier shell correlation between two independent half-sets using the 0.143 threshold[53] and ResMap[54], respectively.

**CryoEM image processing of the partially-enveloped capsid.** The procedures of icosahedral reconstruction and sub-particle refinement of the partially-enveloped capsid were performed as above described. The final maps of the C5 portal vertex, C12 portal, C1 portal vertex, C6 portal vertex, and C1 capsid were determined at global resolutions of 4.0 Å (42,849 particles), 4.2 Å (40,903 particles), 4.8 Å (40,903 particles), 5.3 Å (40,903 particles), and 6.3 Å (40,903 particles), respectively.

**Geometry-based penton vertex sorting.** The penton vertex sub-particles generated from the portal vertex-isolated 3D classification above were responsible for 92.1% of the vertex sub-particles (Supplementary Fig. 2). To determine the CVSC occupancy on the penton vertices in HCMV, we refined the structure of the penton vertex to obtain a 3.6 Å resolution map (C5), then expanded the sub-particle dataset with 5-fold symmetry and performed a round of focused 3D classification

(--tau = 40) with a small mask covering only one CVSC. One of the six converged classes (13.6%, termed as CVSC-binding registers) showed CVSC densities comparable with its surrounding capsid proteins, whereas the remaining five classed clearly lacked CVSC densities (referred to as CVSC-absent registers). The CVSC can bind any of the 5 equivalent registers at the capsid penton vertices, and can theoretically generate 8 types of penton vertices: zero-, one-, four-, and five-CATC-binding vertices; ortho- and meta-CATC-binding vertices (two CATCs bound); and ortho- and meta-CATC-ab sent vertices (three CATCs bound)[36] (Supplementary Fig. 2). First, we readily classified the zero- to five-CATC-binding vertices based on the presence (CVSC-binding register) or absence (CVSC-absent register) of CVSC at one penton vertex. Following this, alternative two- and three-CVSC-binding (two-CVSC-absent) vertices could be isolated according to the value difference of _rlnAngleRot between the two registers (CVSC-binding or CVSC-absent) at one penton vertex: If the angle difference is 72° or −72°, the corresponding penton vertex is an ortho-CVSC-binding or ortho-CVSC-absent vertex; yet, if the angle difference is 144° or −144°, then the corresponding penton vertex is a meta-CVSC-binding or meta-CVSC-absent vertex. Using this strategy, groups of penton vertices in different CVSC-binding geometries were generated, with the sub-particle numbers calculated as 128,702 (zero-CVSC-binding), 131,384 (one-CVSC-binding), 13,783 (meta-CVSC-binding), 11,870 (ortho-CVSC-binding), 1012 (ortho-CVSC-absent), 1083 (meta-CVSC-absent), 96 (four-CVSC-binding), and 5 (five-CVSC-binding), respectively. The final reconstruction of the one-CVSC-binding penton vertex (C1) was determined at a resolution of 4.0 Å.

**Model building**. To build atomic models of the portal vertex and CVSC-binding penton vertex, we fitted the models of the peripenton capsid proteins of HCMV (PDB ID: 5VKU)[24], including the SCP-bound hexon MCPs (P1, P2, P5 and P6), the penton MCP (only for CVSC-binding penton vertex), the triplexes Ta and Tc, and pp150, into our C5 reconstruction of the portal vertex and C1 reconstruction of the CVSC-binding penton vertex with Chimera[55], respectively. The models of the capsid proteins were then manually adjusted in COOT. In addition, most of the CVSC reconstruction revealed well-resolved helix grooves and side-chains for pUL93 and its tightly associated pUL77 conformers; this enabled us to ab initio build their models with the help of secondary structure predictions from *Phyre2*[56]. For modeling the low-resolution part of the CVSC where only smooth helical densities were revealed, including a helix of the upper pUL77 and two isolated short helices of pUL48, we built a homology model based on the structures of the counterparts of HSV-1 CVSC (PDB: 6CGR)[26]. Finally, the models of the capsid proteins and the CVSC in the portal vertex and the CVSC-binding penton vertex were combined and refined together with PHENIX[57].

The atomic model of the aileron domain and the main body of the portal protein pUL104 were built and refined against the C5 map of the portal vertex and the C12 map of the portal, respectively. For modeling of the turret region—owing to the moderate resolution of the density map—we fitted the portal main body into the C6 reconstruction of the portal vertex, and then traced and built the Cα models for the turret regions of the two pUL104 conformers Finally, the model of the two pUL104 conformers, including the portal main body and the turret region, were refined against the C6 map in PHENIX[57].

**Reporting summary**. Further information on research design is available in the Nature Research Reporting Summary linked to this article.

## Data availability

All density maps have been deposited in the Electron Microscopy Bank under accession codes EMD-31297 (C5 portal vertex from the intact virion), EMD-31295 (C12 portal from the intact virion), EMD-31292 (C1 capsid from the intact virion), EMD-31290 (C1 portal vertex from the intact virion), EMD-31299 (C6 portal vertex from the intact virion), EMD-31301 (C1 one CVSC-binding penton vertex from the intact virion), EMD-31298 (C5 portal vertex from the partially-enveloped nucleocapsid), EMD-31296 (C12 portal from the partially-enveloped nucleocapsid), EMD-31293 (C1 capsid from the partially-enveloped nucleocapsid), EMD-31291 (C1 portal vertex from the partially-enveloped nucleocapsid) and EMD-31300 (C6 portal vertex from the partially-enveloped nucleocapsid). The atomic coordinates have been deposited in the Protein Data Bank under accession code 7ET2 [https://doi.org/10.2210/pdb7ET2/pdb] (C12 portal), 7ETM [https://doi.org/10.2210/pdb7ETM/pdb] (C6 portal), 7ET3 [https://doi.org/10.2210/pdb7ET3/pdb] (C5 portal vertex from the intact virion), 7ETJ [https://doi.org/10.2210/pdb7ETJ/pdb] (C5 portal vertex from the partially-enveloped nucleocapsid) and 7ETO [https://doi.org/10.2210/pdb7ETO/pdb] (C1 one CVSC-binding penton vertex).

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

## Acknowledgements

The cryo-EM data were collected at Cryo-Electron Microscopy Research Center, Shanghai Institute of Material Medica. This work was partially supported by the 100 Talents Program of the Chinese Academy of Sciences (to X.Y.); Natural Science Foundation of Shanghai (18ZR1447700 to X.Y.); the National Natural Science Foundation of China (31900869 to Z.L.); Shanghai Sailing Program (19YF1456800 to Z.L.).

## Author contributions

X.Y. conceived the project, and designed and supervised the research. J.P. and L.D. prepared the samples. Z.L. and J.P. collected the data, determined the structures, and prepared the figures. Z.L. and L.D. built the models. X.Y. and Z.L. interpreted the results. X.Y. wrote the manuscript. All authors reviewed and edited the manuscript.

## Competing interests

The authors declare no competing interests.
