## [Peer Review File · Nature Communications]

REVIEWER COMMENTS

Reviewer #1 (Remarks to the Author):

This paper describes an impressive structural study using cryoEM on HCMV; a betaherpesvirus and pervasive human pathogen. I am not an expert in herpesvirus biology, but have significant experience in structural virology - the quality of the structural work seems high, and I am confident the result is of significant biological interest to a general readership, especially the details of the C5 portal turret, and I found many features of the capsid – e.g. the asymmetric patterns of CVSC binding – fascinating. I recommend publication of the work.

However, some significant changes to the manuscript should be made to increase its accessibility – this is structural description of a very large and complicated piece of biology and at the moment I found it to be very difficult to engage with – suggesting that a general audience might find it impossible.

1. The MS has a significant number of minor errors that make it more difficult to read. I would suggest that some professional editorial input would enhance the accessibility.
2. In similar vein, there are some sentences that are barriers to understanding rather than helpful. A good example is in the Abstract, “Through contacting with the portal-encircling DNA, the 10-helix anchor, which is not found in other known portals, likely function as a damper for the portal reaching the headful state, thus to lead the large genome of HCMV to be packaged; the turret, with one of its 6 coiled coils latched by a helix from a major capsid protein, lend support to the portal to retain the packaged genome; the CVSCs, presumably increasing the inner capsid pressure, low-stoichiometrically occupy the penton vertices to facilitate the large genome’s retention.”. I would suggest this should be broken into at least three sentences, each with a clear message.
3. The authors should be wary about stating as fact the discussive points of their MS, eg. “is specifically adapted to pack its enormous genome.”
4. As a minor technical point, it’s a nice result that their classification identifies 7.9% of their sub-volumes as containing a portal – based on 1:12 this figure should be 8.3%. Does the small discrepancy reflect defective particles or limitations in data quality?
5. I am not surprised that the structure of intact HCMV capsid is limited to around 4 Å, but I am little surprised that when they relax symmetry, and e.g. apply 12-fold symmetry to the portal the resolution falls rather than rises. Clearly an offset between local/global order and number of particles is happening. I wonder if the authors could comment on the source of the resolution limit in their reconstructions?
6. I have a strong preference for avoiding statements like ‘obviously’ (line 146) in connection with complex structural features that the authors are intimately familiar with and the audience is not – these should be shown/explained.
7. I think the figures are a major issue for this MS – Firstly, 5 main and 16 extended figure seems like the wrong balance. It has lead to large and complicated compound figures that are very difficult to follow.

As an example, Fig1 is too complex, I don’t readily understand the flow of views, or colour schemes that identify components and interactions. We have dashed boxes and arrows that aren’t clear, in different colours that don’t relate to the same colours used in the structural components, and labels that are too small and will be illegible at the scale viewed by a reader, and in colours (and overlaid on structure) that make it very difficult.

The legend is very terse too. Perhaps to meet the <350 word limit when there is so much going on?

I have no easy answers for this – but I think the paper needs to grow to fill more space, and take more time to explain the beautiful result. Nature Comms has (I believe?) a guideline of 10 display items. I think all of these will be needed!

I'd be very happy to look at this beautiful structure again in a revised MS.

Reviewer #2 (Remarks to the Author):

The manuscript “Structural basis for genome package, retention and ejection in human cytomegalovirus” by Li et al reports on the cryoEM structure of the cytomegalovirus virion capsid. The claim is of 4.0Å resolution, and this would represent an advance over previous cryoEM structures of HCMV. The herpesvirus capsid is very complex with symmetrically disposed viral structural proteins in hundreds of copies that make up the capsid studied here. The cryoEM used a high-end microscope, and a direct electron-detecting camera, so in principle data quality should be high. Although, I do not have the expertise (CryoEM) to critically evaluate the structural data with regards to the details of the cryoEM reconstructions. The manuscript lists a number of architectural findings of the portal and capsid vertex-specific components (CVSC) – in particular the structure of the HCMV portal consisting of a C12 main body, C5 10-helix anchor and the C6 turret. The C5 10-helix anchor and C6 turret appear to be unique to the HCMV portal as compared to other herpesvirus (HSV, KSHV, EBV) virion portals. The authors proposed that both of these portal components function to anchor the large HCMV viral genome in the capsid via the C5 10-helix interacting with the packaged DNA and the C6 turrets interaction with the major capsid protein. In addition, the copy number of the CVSC on each HCMV virion capsid was found to be less compared to other herpesvirus virion capsids and propose that this functions to allow packaging and retention of the HCMV genome that is larger than other herpesvirus genomes. Finally, structural studies of partially de-enveloped virions demonstrated that the portal undergoes conformational changes that may be required for the subsequent release of the viral genome.

There are a number of other high-resolution cryoEM reconstructions of herpesvirus portal structures. The results described here constitute a significant advance in our knowledge of the possible role of the portal in genome packaging, retention and ejection. The structure of the HCMV portal shares common features with other herpesviruses but also differences that may contribute the packaging of the much larger HCMV genome. The writing can be improved with many errors in grammar that should be addressed. Otherwise, this is a very nice report filled with intriguing findings and should be of interest to investigators studying virus structure and to people interested in targeting virus assembly for development of novel antiviral agents. I only have a few suggestions.

1. Extended Fig. 16. Can the missing turret helices for HSV, EBV, and KSHV be found in the difference in the amino acid sequence of these proteins compared the HCMV? If so a comparison of the protein sequences of the different portal proteins should be added to this figure.

2. Line 196 and Discussion. Is the resolution high enough to determine at least some of the sequence of the 148-bp pDNA that encircles the crown region? Would predict that this

should be the S-end of the genome since it is predicted to be the last to enter the capsid and first to leave.

3. Discussion and last section of results. What happens to the portal cap (Fig. 1B) when the viral envelope is extracted? The cap is predicted to be part of the C-terminal region of pUL77 that is required for the stable packaging of the viral genome. Removing the envelope shouldn't alter the cap structure.

Reviewer #3 (Remarks to the Author):

The authors describe the reconstruction of the HCMV nucleocapsid within the context of the virions. The reconstructions are well done and most of the description is reasonable. Of particular interest is the identification of parts of the portal protein not seen in reconstructions of related viruses. The weakness of the manuscript is the modeling of the vertex-associated proteins (CVSC) as discussed below. I do believe the manuscript needs significant revision before being suitable for publication.

The resolution is impressive for ~20k particles. However, it seems that one of the resolution-limiting factors is the pixel size of ~1.6 Å, which realistically limits the resolution to ~4.8 Å. Also, the helices appear very smooth, which indicates that the resolution is closer to ~6 Å. Together, I don't think the reconstructions are actually better than 6 Å. One problem could be the mask used in calculating the FSC curves. This is not mentioned in the methods and I presume they just used the default in Relion. At a ~6 Å resolution, structural refinements do not mean much and scores such as the Molprobit score is unrealistic. In particular, the ab initio tracing of the CVSC structures is highly suspect and cannot be supported with the data presented. The discussion of these modeled structures are therefore not believable.

We would like to express our gratitude to the editor and the three reviewers for spending time in evaluating our paper. As you will see from our point-by-point response below, we have been thorough in our attempts to address all the points raised by the reviewers. We have prepared additional figures and revised the paper accordingly.

To facilitate your navigation of this response statement, the referees' comments are in **black**, and our responses are in **blue**.

Referees' comments:

Reviewer #1 (Remarks to the Author):

This paper describes an impressive structural study using cryoEM on HCMV; a betaherpesvirus and pervasive human pathogen. I am not an expert in herpesvirus biology, but have significant experience in structural virology - the quality of the structural work seems high, and I am confident the result is of significant biological interest to a general readership, especially the details of the C5 portal turret, and I found many features of the capsid – e.g. the asymmetric patterns of CVSC binding – fascinating. I recommend publication of the work.

Re: We are very grateful for the reviewer's positive comments on our study.

However, some significant changes to the manuscript should be made to increase its accessibility – this is structural description of a very large and complicated piece of biology and at the moment I found it to be very difficult to engage with – suggesting that a general audience might find it impossible.

1. The MS has a significant number of minor errors that make it more difficult to read. I would suggest that some professional editorial input would enhance the accessibility.

Re: We apologize for these errors in the original manuscript. Per the reviewer's suggestions, we have had our manuscript reviewed by a professional editing service and corrected all of the grammatical and typographical errors in the manuscript.

2. In similar vein, there are some sentences that are barriers to understanding rather than helpful. A good example is in the Abstract, "Through contacting with the portal-encircling DNA, the 10-helix anchor, which is not found in other known portals, likely function as a damper for the portal reaching the headful state, thus to lead the large genome of HCMV to be packaged; the turret, with one of its 6 coiled coils latched by a helix from a major capsid protein, lend support to the portal to retain the packaged genome; the CVSCs, presumably increasing the inner capsid pressure, low-stoichiometrically occupy the penton vertices to facilitate the large genome's retention." I would suggest this should be broken into at least three sentences, each with a clear message.

Re: The whole manuscript including the Abstract has been re-edited to make it more accessible for a general readership.

3. The authors should be wary about stating as fact the discussive points of their MS, eg. "is specifically adapted to pack its enormous genome."

Re: We agree and have revised this sentence.

4. As a minor technical point, it's a nice result that their classification identifies 7.9% of their sub-volumes as containing a portal – based on 1:12 this figure should be 8.3%. Does the small discrepancy reflect defective particles or limitations in data quality?

Re: It is possible that a few of the assembled empty viral particles contain no portal because, in the absence of portal, capsid proteins can still assemble into a capsid. However, the particles used in this study for the final structure determination were manually picked to ensure that all particles were DNA-containing particles with portals. We thus believe that the slightly lower sub-particle ratio of the portal vertex as compared with the theoretical number (8.3%) was caused by limitations in image quality; i.e., ice contamination or a low signal contrast of the images.

5. I am not surprised that the structure of intact HCMV capsid is limited to around 4 Å, but I am little surprised that when they relax symmetry, and e.g. apply 12-fold symmetry to the portal the resolution falls rather than rises. Clearly an offset

between local/global order and number of particles is happening. I wonder if the authors could comment on the source of the resolution limit in their reconstructions?

Re: We thank the reviewer for pointing out the technical issue on the map resolution assessment. We think the major reason for the lower resolution of the C12 portal is that the tightly associated DNA around the portal impaired a high-resolution structural determination of the C12 portal during the iterative refinement. In addition, those DNAs are included in the final resolution determination of the C12 portal and thus would likely further decrease the overall resolution of the reconstruction.

6. I have a strong preference for avoiding statements like 'obviously' (line 146) in connection with complex structural features that the authors are intimately familiar with and the audience is not – these should be shown/explained.

Re: These sentences have been revised.

7. I think the figures are a major issue for this MS – Firstly, 5 main and 16 extended figure seems like the wrong balance. It has lead to large and complicated compound figures that are very difficult to follow.

Re: Following the reviewer's suggestion, we have adjusted the balance between the main and extended figures in the revised manuscript. We now present 9 main figures and 14 extended figures, and we have reduced the complexity of the figures.

As an example, Fig1 is too complex, I don't readily understand the flow of views, or colour schemes that identify components and interactions. We have dashed boxes and arrows that aren't clear, in different colours that don't relate to the same colours used in the structural components, and labels that are too small and will be illegible at the scale viewed by a reader, and in colours (and overlaid on structure) that make it very difficult.

Re: We apologize for the confusion. We have modified the corresponding figures to make them easier to follow in the revised manuscript.

The legend is very terse too. Perhaps to meet the <350 word limit when there is so much going on?

Re: The legends have been revised to describe our results more clearly in the revised manuscript.

I have no easy answers for this – but I think the paper needs to grow to fill more space, and take more time to explain the beautiful result. Nature Comms has (I believe?) a guideline of 10 display items. I think all of these will be needed!

Re: As mentioned above, we have increased the number of main figures to 9, and reworked the figures so that they are easier to follow. We have also expanded the Results section (lines 191-193 and lines 340-342).

I'd be very happy to look at this beautiful structure again in a revised MS.

Thank you very much. We hope that we have sufficiently addressed your concerns.

Reviewer #2 (Remarks to the Author):

The manuscript "Structural basis for genome package, retention and ejection in human cytomegalovirus" by Li et al reports on the cryoEM structure of the cytomegalovirus virion capsid. The claim is of 4.0Å resolution, and this would represent an advance over previous cryoEM structures of HCMV. The herpesvirus capsid is very complex with symmetrically disposed viral structural proteins in hundreds of copies that make up the capsid studied here. The cryoEM used a high-end microscope, and a direct electron-detecting camera, so in principle data quality should be high. Although, I do not have the expertise (CryoEM) to critically evaluate the structural data with regards to the details of the cryoEM reconstructions. The manuscript lists a number of architectural findings of the portal and capsid vertex-specific components (CVSC) – in particular the structure of the HCMV portal consisting of a C12 main body, C5 10-helix anchor and the C6 turret. The C5 10-helix anchor and C6 turret appear to be unique to the HCMV portal as compared to other herpesvirus (HSV, KSHV, EBV) virion portals. The authors proposed that both of these portal components function to anchor the large HCMV viral genome in the capsid via the C5 10-helix interacting with the the packaged DNA and the C6 turrets interaction with the major capsid protein. In addition, the copy number of the CVSC on each HCMV virion capsid was found to be less compared

to other herpesvirus virion capsids and propose that this functions to allow packaging and retention of the HCMV genome that is larger than other herpesvirus genomes. Finally, structural studies of partially de-enveloped virions demonstrated that the portal undergoes conformational changes that may be required for the subsequent release of the viral genome. There are a number of other high-resolution cryoEM reconstructions of herpesvirus portal structures. The results described here constitute a significant advance in our knowledge of the possible role of the portal in genome packaging, retention and ejection. The structure of the HCMV portal shares common features with other herpesviruses but also differences that may contribute the packaging of the much larger HCMV genome. The writing can be improved with many errors in grammar that should be addressed. Otherwise, this is a very nice report filled with intriguing findings and should be of interest to investigators studying virus structure and to people interested in targeting virus assembly for development of novel antiviral agents. I only have a few suggestions.

Re: We thank the reviewer for the positive comment on our work. We have gone through the whole manuscript and tried our best to correct the grammatical errors in the revised manuscript. We have also had our manuscript reviewed by a professional editing service.

1. Extended Fig. 16. Can the missing turret helices for HSV, EBV, and KSHV be found in the difference in the amino acid sequence of these proteins compared the HCMV? If so a comparison of the protein sequences of the different portal proteins should be added to this figure.

Re: We apologize for any misunderstanding regarding Extended Fig. 16, which was used to show the unique 6-fold symmetric assembly of the HCMV portal turret. The reconstructions of other three herpesviruses also revealed helical densities for the portal turret, but these were all arranged in a 5-fold symmetry and were unable to be modeled due to the symmetry-mismatch with their corresponding portal main bodies (Liu, et al, Nature 2019; Gong, et al, Cell 2019; Li, et al, Cell Research 2020). We have modified the corresponding figure (Extended Data Fig. 16 in the original manuscript) in the revised manuscript (Extended Data Fig. 14).

2. Line 196 and Discussion. Is the resolution high enough to determine at least some of the sequence of the 148-bp pDNA that encircles the crown region? Would predict that this should be the S-end of the genome since it is predicted to be the last to enter the capsid and first to leave.

Re: The resolution for pDNA in this study is not high enough to determine its sequence. The last-to-enter terminal segment of the genome (the terminal DNA) was held in the DNA translocation channel (Fig. 2b), as also proposed by a previous study of HSV-1 (Liu, et al, Nature 2019). The pDNA is proposed to be the first-packaged genomic segment (Ray, K., et al, J. Mol. Biol. 2010; Liu, et al, Nature 2019), and thus it is unlikely to be the S-end of the genome.

3. Discussion and last section of results. What happens to the portal cap (Fig. 1B) when the viral envelope is extracted? The cap is predicted to be part of the C-terminal region of pUL77 that is required for the stable packaging of the viral genome. Removing the envelope shouldn't alter the cap structure.

Re: The reviewer's prediction is correct. The portal caps of enveloped and partially-enveloped capsids are essentially identical. We have included a comparison of the portal cap in the Results section in the revised manuscript (lines 340-342). We thank the reviewer for the advice.

Reviewer #3 (Remarks to the Author):

The authors describe the reconstruction of the HCMV nucleocapsid within the context of the virions. The reconstructions are well done and most of the description is reasonable. Of particular interest is the identification of parts of the portal protein not seen in reconstructions of related viruses. The weakness of the manuscript is the modeling of the vertex-associated proteins (CVSC) as discussed below. I do believe the manuscript needs significant revision before being suitable for publication.

The resolution is impressive for ~20k particles. However, it seems that one of the resolution-limiting factors is the pixel size of ~1.6 Å, which realistically limits the resolution to ~4.8 Å. Also, the helices appear very smooth, which indicates that the resolution is closer to ~6 Å. Together, I don't think the reconstructions are actually better than 6 Å. One problem could

be the mask used in calculating the FSC curves. This is not mentioned in the methods and I presume they just used the default in Relion. At a ~ 6 Å resolution, structural refinements do not mean much and scores such as the Molprobity score is unrealistic. In particular, the ab initio tracing of the CVSC structures is highly suspect and cannot be supported with the data presented. The discussion of these modeled structures are therefore not believable.

Re: We thank the reviewer for these comments.

We calculated the global resolution of the reconstruction of the portal vertex using the mask that contains all of the components, including the CVSC complex. As illustrated by the local resolution distribution (Supplementary Fig. 3), the helix bundle of the CVSC was the lowest-resolution region. Thus, to show the density continuity of the CVSC, we prepared certain figures (Figs. 1b, 1c, 1f and 4d, Extended Data Figs. 10 and 11 in the original manuscript) using the unsharpened maps of portal/penton vertex, which indeed show smooth helices. We apologize for the confusion and we have indicated the map information in the revised figure legends. Actually, the B-factor-sharpened reconstruction of most of the CVSCs, which was shown in Fig. 4 and Extended Data Fig. 4b in the original manuscript, revealed well-resolved helix grooves and side-chain features for the proteins of pUL93 and its tightly associated pUL77 conformers, and is sufficient for the ab initio build of their models. For modeling the low-resolution part, including the helix of the upper pUL77 and the two isolated short helices of pUL48, we built a homology model based on the structures of these counterparts from HSV-1 (Dai, Science 2018). We have included a detailed description of the CVSC modeling procedure in the revised manuscript (lines 527-533).

It is true that, in the early days of the cryoEM technology with photographic film or the charge-coupled device (CCD), the realistic target resolution of a cryoEM structure was around 3-times the pixel size or the 2/3 Nyquist due to the poor detective quantum efficiency (DQE). In recent years, however, this has been changed with the application of the direct electron detection device (DDD), which has novel features: i.e. better DQE, recording cryo-EM data as dose-fractionated image stacks instead of as a single micrograph, and a more effective way to deal with rapid radiation damage. The DDD and its features enable structure determination of biological molecules at resolutions higher than 3-times the pixel size, even close to the Nyquist (2-times the pixel size) (examples can be found in Yin, et al, Science 2020 and Pintilie et al, Nat. Methods 2020).

REVIEWER COMMENTS

Reviewer #1 (Remarks to the Author):

The authors have done a great job in revising the manuscript - the figures are very significantly improved in style and clarity (and number!), and the revised text now takes the reader through the structure in a much more accessible way. I'm happy with the technical aspects of the structure determination and image processing and am happy to recommend the manuscript for publication.

I would still recommend some minor amendments to the figures - e.g. in Figure 1b, I understand the rationale for green text for "Portal Cap" when the density is green, but having the green text on top of green density is not very accessible. Throughout there are tweaks that could enhance the contrast of text labels... The pinkish text in Fig 8a is a prime example!

Reviewer #3 (Remarks to the Author):

The authors clarified a number of points and reworked the figures to show more convincing features closer to the 4 Å estimated resolution. Overall the manuscript is much improved. I have a number of minor points that need to be addressed.

Line 93 & line 388:

It is not clear how the lower stoichiometry of CVSCs would be beneficial for retention. The logical conclusion would be that expulsion would be more likely with fewer CVSCs. The authors mention a relationship between the numbers of bound CVSC and genome size and refer to their prior work. However, it is a very speculative theory and not based on actual estimates of pressure and how the CVSCs affect this. For instance, an alternative theory could hold that it is the strength of interactions of the triplexes and the capsomer floor domains that determine how much pressure the capsid can withstand. The detail around the portal interactions may similarly be important for retention/ejection of the DNA and not CVSCs. Until such issues are clarified I would regard the reason for the CVSC stoichiometry as unresolved.

Lines 252-260:

The terms the authors invented to describe the CVSC patterns should be explained. The reader does not intuitively know what "ortho" and "meta" mean in this context. The description in the Methods section (lines 496-524) is too late. While it is shown in supplementary figure 2, it is still confusing and it requires the reader to look at the figure to get a sense of what it means.

Line 435:

It is not clear in what mode the K3 camera was operated (linear, counting, superresolution).

Line 450:

What was the resolution obtained before Ewald sphere correction?

Line 482:

Typo: "reconstrued"

Line 486:

"Golden-standard"

I would prefer "independent half-sets".

The legend for Supplementary figure 2 should include a brief description of the workflow. At a minimum the different colors should be explained. For instance, does the red-green-blue color scheme represent local resolution? In the CVSC-binding pattern cases, what do the colors indicate and why are two diagrams and the others reconstructions?

We would like to express our gratitude to the three reviewers for spending time in evaluating our paper and for recognizing the merit of our work. We also thank the editor for supporting us throughout this process.

To facilitate your navigation of this response statement, the referees' comments are in **black**, and our responses are in **blue**.

Referees' comments:

Reviewer #1 (Remarks to the Author):

The authors have done a great job in revising the manuscript - the figures are very significantly improved in style and clarity (and number!), and the revised text now takes the reader through the structure in a much more accessible way. I'm happy with the technical aspects of the structure determination and image processing and am happy to recommend the manuscript for publication.

Re: We thank the reviewer for the support of our work.

I would still recommend some minor amendments to the figures - e.g. in Figure 1b, I understand the rationale for green text for "Portal Cap" when the density is green, but having the green text on top of green density is not very accessible. Throughout there are tweaks that could enhance the contrast of text labels... The pinkish text in Fig 8a is a prime example!

Re: The figure has been revised and the text label mentioned by the reviewer has been enhanced.

Reviewer #3 (Remarks to the Author):

The authors clarified a number of points and reworked the figures to show more convincing features closer to the 4 Å estimated resolution. Overall the manuscript is much improved. I have a number of minor points that need to be addressed.

Line 93 & line 388:

It is not clear how the lower stoichiometry of CVSCs would be beneficial for retention. The logical conclusion would be that expulsion would be more likely with fewer CVSCs. The authors mention a relationship between the numbers of bound CVSC and genome size and refer to their prior work. However, it is a very speculative theory and not based on actual estimates of pressure and how the CVSCs affect this. For instance, an alternative theory could hold that it is the strength of interactions of the triplexes and the capsomer floor domains that determine how much pressure the capsid can withstand. The detail around the portal interactions may similarly be important for retention/ejection of the DNA and not CVSCs. Until such issues are clarified I would regard the reason for the CVSC stoichiometry as unresolved.

Re: We agree with the reviewer on that the strength of interactions among capsid proteins play the primary role for retention/ejection of the DNA. However, based on previously published data and our study here, we believe the CVSCs could benefit the DNA's retention/ejection.

The portal vertex has been suggested to be the weakest point of the capsid (D. W. Bauer et al, JVI 2015), thus the pressure imposing to the portal or the portal vertex should be precisely modulated to benefit both the genome's retention and ejection. Previous structural analysis of nucleocapsid from different herpesviruses have consistently revealed that the penton vertices are the most flexible, which would conceivably relieve the enormous inner capsid pressure (Yu et al,

Science 2017; Dai and Zhou, Nature 2018; Dai et al, Science 2018; Li et al, Cell Res 2020). The CVSC, through binding to the penton-vertex registers in a stoichiometric manner, could reduce the mechanical elasticity of those capsid penton vertices, which in turn would increase the inner capsid pressure; this speculation is further supported by the fact that the CVSC copy number on capsid penton vertices is inversely related to the genome sizes among herpesviruses.

Lines 252-260:

The terms the authors invented to describe the CVSC patterns should be explained. The reader does not intuitively know what "ortho" and "meta" mean in this context. The description in the Methods section (lines 496-524) is too late. While it is shown in supplementary figure 2, it is still confusing and it requires the reader to look at the figure to get a sense of what it means.

Re: Following the reviewer's suggestion, we have added the description of the terms "ortho" and "meta" where they show up for the first time (lines 253-254).

Line 435:

It is not clear in what mode the K3 camera was operated (linear, counting, superresolution).

Re: We collected the cryoEM images with K3 camera in super-resolution mode and this information has been added in the revised manuscript (line 432).

Line 450:

What was the resolution obtained before Ewald sphere correction?

Re: The icosahedral reconstruction of HCMV virion capsid was determined at 4.4 Å before Ewald-sphere curvature correction. This information has been added in the revised manuscript (line 447-449)

Line 482:

Typo: "reconstrued"

Re: It has been corrected.

Line 486:

"Golden-standard"

I would prefer "independent half-sets".

Re: Following the reviewer's suggestion, the "Golden-standard Fourier shell correlation" has been replaced with "Fourier shell correlation between two independent half-sets".

The legend for Supplementary figure 2 should include a brief description of the workflow. At a minimum the different colors should be explained. For instance, does the red-green-blue color scheme represent local resolution? In the CVSC-binding pattern cases, what do the colors indicate and why are two diagrams and the others reconstructions?

Re: Following the reviewer's suggestion, we have added a brief description of the workflow in the legend to make it easier to follow.